# Relationship between the Dietary Inflammatory Index and Cardiovascular Health among Children

**DOI:** 10.3390/ijerph192315706

**Published:** 2022-11-25

**Authors:** Ana Isabel Mora-Urda, Francisco Javier Martín-Almena, María del Pilar Montero López

**Affiliations:** 1Departamento de Didácticas Específicas, Facultad de Formación del Profesorado, Universidad Autónoma de Madrid, 28049 Madrid, Spain; 2Facultad de Ciencias de la Salud, Universidad Católica de Ávila, 05005 Avila, Spain; 3Departamento de Biología, Facultad de Ciencias, Universidad Autónoma de Madrid, 28049 Madrid, Spain

**Keywords:** dietary inflammatory index, cardiovascular health, children, blood pressure, body fat mass, lifestyles, KIDMED, 24 h recall surveys

## Abstract

Background: The aim of this study was to evaluate the relationship between the Dietary Inflammatory Index (DII^®^) and cardiovascular health indicators in children. Methods: The sample consisted of 365 schoolchildren aged 8 to 12 from the Region of Madrid. Anthropometric and hemodynamic measurements were collected. Variables relating to habits and lifestyles, parental level of education, and data on their diet, through three 24 h food recall surveys, were also collected. The diet quality indicators considered are the DII based on 25 nutrients and the KIDMED index. Results: Children with a more pro-inflammatory diet came from families with lower levels of parental education (*p* < 0.05). Predictive models show that in the group with a more pro-inflammatory diet (>P50), the likelihood of developing hypertension in childhood is 2.1 times higher (OR = 2.085 (1.107–3.927)) and they have more than twice the risk of developing obesity (OR = 2.3) or developing obesity and hypertension simultaneously (OR = 1.290 (1.316–3.985)). Furthermore, predictive models showed that the children with a pro-inflammatory diet (>P50) had higher values for BFM% (β = 1.957; *p* = 0.026) and BMI (β = 0.015; *p* = 0.012) than children with a lower inflammatory diet (<P50). Conclusions: Higher values on the DII are related to poorer nutritional status and cardiovascular health in childhood. Thus, a pro-inflammatory diet is also associated with a lower socio-economic level and poorer diet quality.

## 1. Introduction

Inflammation is a result of the body’s response to tissue insult or injury, or the presence of inflammatory stimulants. Chronic inflammation is known to be associated with cardiovascular disease (CVD), which is the leading cause of death and accounts for about half of adult deaths [1]. Furthermore, there is also evidence that diet and nutrition play a central role in regulating inflammation processes [2]. In order to quantify the inflammatory potential of diet, the Dietary Inflammatory Index (DII) was developed according to the pro- and anti-inflammatory properties of the nutrients that make up food [3,4]. It was created and validated in a longitudinal study that assessed changes in C-reactive protein (CRP) levels, depending on the frequency of consumption of certain foods [3,4]. CRP is produced in response to interleukin (IL) stimulation, such as IL-6 [5]. The development of high-sensitivity tests for C-reactive protein (hs-CRP) has made it possible to detect inflammation at a vascular level [6]. The high clinical specificity shown by hs-CRP explains its frequent use as a target for analysis over other inflammatory markers [7].

Diets with anti-inflammatory potential are associated with low scores on this in-dex, i.e., reduced inflammatory states and the important relationship between the different stagesof life, as childhood, and the development of diseases in adulthood [8]. This is because some nutrients present in various foods can activate molecules and signalling pathways of inflammatory processes [9].

One recent hypothesis is that obesity may also be partly a consequence of previous chronic low-grade inflammation; therefore, there may be an association between inflammation and obesity [10,11]. Adipose tissue is recognised as an active regulator of pathological processes such as immunity and inflammation. This tissue is responsible for the production and release of adipokines and pro-inflammatory cytokines, as it contains macrophages and adipocytes that release these molecules [12]. Interleukin IL-1 and IL-6, produced by adipose tissue, are present in high concentrations in obese individuals with insulin resistance, so among other factors (genetic, biological, sociocultural…), a state of chronic inflammation can contribute to the onset and development of disease, atherosclerosis in people with obesity [13].

With regard to hypertension, some studies show that subtle differences in the composition of the diet, due to the pro-inflammatory or anti-inflammatory nature of the nutrients, significantly alter the association with markers of atherosclerosis [14]. A diet with high levels of fat consumption, especially saturated fat, and low fibre and micronutrient consumption, is associated with an increase in the presence of markers for atherosclerosis and joint stiffness. Hypertension is one of the greatest indicators of joint stiffness, and several studies show that hypertensive children have an increase in the intima-medial thickness (IMT) of the carotid artery [15]. Ultimately, adipose and vascular tissue regulate inflammatory processes, directly influencing the pathogenesis of chronic diseases [12]. A constant inflammatory state leads to serious metabolic dysfunction, contributing to the appearance and development of obesity and endothelial dysfunction. The combination of these pathological mechanisms can lead to severe diseases such as atherosclerosis or metabolic syndrome, promoting vascular and thrombotic lesions responsible for CVD [13,16,17].

Therefore, it may be useful to characterise an individual’s diet based on its inflammatory properties to investigate the inflammatory links between obesity and diet [18]. The Dietary Inflammatory Index (DII) is a new tool for assessing this inflammatory potential of diet [4]. In the present study, whilst controlling for socio-economic status and lifestyle factors (physical activity and screen time), we assess the relationships between nutrient intake or food group consumption and the DII, as well as the association between the DII and cardiovascular health indicators (central and abdominal obesity and blood pressure) in a sample of 8–12-year-old schoolchildren living in the Region of Madrid. 

## 2. Materials and Methods

### 2.1. Population and Sample Group Characteristics

Data were gathered at schools in the Autonomous Region of Madrid, having first gained the approval of the school board, in accordance with the ethical standards for research on human beings set forth in the Helsinki Protocol [19] and Spanish Organic Law 15/1999 of 13 December on personal data protection. The project protocol was approved by the ethics committee of the Universidad Autónoma de Madrid (REF: CEI-29-716). Participation in the study was voluntary, and the families all signed informed consent documents allowing their children to take part, prior to the collection of any data. Only children with no previous history of arterial disease or hypertension who were not currently taking any anti-hypertension medication were included in the study; girls experiencing the onset of menarche were also excluded. Girls with menarche were excluded due to changes in body composition. 

The sample consisted of 365 children (190 boys and 175 girls) aged 8–12, with an average age of 9.7 (sd = 1.1) for boys and 9.6 (sd = 1.2) for girls (t = 13112; *p* = 0.267). 

### 2.2. Socio-Demographic Variables, Parental Phenotypes, Peri- and Postnatal Factors

Parental socio-economic and anthropometric characteristics were obtained through interviews with families, including maternal/parental education (higher education and others). 

### 2.3. Anthropometric Variables and Indexes in Childhood

Weight (kg) and height (cm) were measured according to the International Biological Program guidelines [20]. All measurements were taken by specially trained staff, using certified and approved instruments. Height was measured with a GPM anthropometer (SiberHegner, Zurich, Switzerland) with a range of 0–2.100 mm and precision of 0.1 mm, and weight was measured on a digital scale with a range of 0–130 kg and precision of 100 g. We also measured umbilical waist circumference (WC, cm) as an indicator of abdominal adiposity [21,22].

Z-scores for weight, height, and BMI were calculated using the 2007 World Health Organization (WHO) values as the reference [23], classifying the children as thin (<−2 SD), normal weight (between −2 SD and 1 SD), overweight (between 1 and 2 SD) and obese (≥2 SD). 

Biceps (mm), triceps (mm), subscapular (mm) and suprailiac (mm) subcutaneous fat skinfolds were measured using “Holtain” skinfold callipers (Holtain, Crymych, UK) with a range of 0–48 mm and precision of 0.2 mm. Body fat mass percentage (BFM%) was calculated using skinfold data after calculating body density (D) using Brook’s equation [24]: Boys: D (kg/cm^3^) = 1.1690–0.0788 × log (Σ skinfolds)Girls: D (kg/cm^3^) = 1.2063–0.0999 × log (Σ skinfolds)

Having obtained D, calculations were made to find the BFM% using the expression proposed by Siri [25]:

BFM% = [(4.95/D) − 4.50] × 100

Fat mass (FM, kg) was calculated as (BFM%/100) × body weight (kg). Fat mass index (FMI, kg/m^2^) was FM (kg) divided by height squared (m^2^), in order to adjust the body composition for height [26].

### 2.4. Arterial Pressure in Childhood 

Systolic and diastolic blood pressure (SBP and DBP, 0–280 mmHg) were measured with a Diagnostec EW-BU30 automatic oscillometric tensiometer (Panasonic, Kadoma, Japan). All measurements were taken in the morning, with the child in a seated position with their back supported and left arm uncovered and supported at heart level, legs uncrossed, and feet flat on the floor after at least 5 min of rest. Arm circumference was measured at the midpoint between the acromion and olecranon in order to select the appropriate cuff (17–30 cm). Two consecutive measurements were registered, with the mean taken as the clinical SBP and DBP. In cases yielding high blood pressure values, the family was informed and a visit to the paediatrician was recommended. The values obtained for each child were compared against the reference values for sex, age and height given in ‘The Fourth Report on the Diagnosis, Evaluation and Treatment of High Blood Pressure in Children and Adolescents” [27]. Subjects with blood pressure values below the standard percentile 95 (p95) were placed in the ‘normotensive’ category, and children with SBP or DBP above the standard percentile 95 (≥p95) were considered ‘hypertensive’. Mean arterial pressure (MAP) was calculated using the formula [SBP + (2 × DBP)]/3. 

### 2.5. Obesity and Hypertension in Childhood

Children with obesity and arterial hypertension (AHT) were identified, and a new variable was created, cardiovascular risk factor (CVRF), comprising two categories: 0. Absence of CVRF and 1. Presence of obesity and hypertension. 

### 2.6. Nutrient Intake and Diet Quality

Quantitative and qualitative information was obtained on the diet of all the children. The variables collected were the number of meals per day and place of the main meal, information concerning the quality of the diet, evaluated by means of the KIDMED Index [28], which includes 16 items; those which reflect unhealthy habits score −1 point and healthy ones score +1. 

Quantitative data on all food consumed over three days were also collected through three 24 h recall surveys (two weekdays and one weekend day). To facilitate the estimation of the quantities of food consumed, the photo album of food portions validated by Hercberg et al. [29] was used. Based on these data, the average energy and nutritional composition of their food intake over these three days was calculated (energy consumption (kcal/day), grams per day of carbohydrates, lipids, proteins, saturated fatty acids (SFA), polyunsaturated fatty acids (PUFA), monounsaturated fatty acids (MUFA), cholesterol, fibre, minerals and vitamins). Different food quality indices were calculated as the percentage of energy consumed from carbohydrates, proteins, and lipids (caloric profile), cholesterol consumption per thousand kilocalories (mg/1000 kcal), and the ratio of MUFA, PUFA and SFA using the equation MUFA + PUFA/SFA. The nutritional composition of the food consumed was analysed individually using the DIAL^®^ programme [30]. 

### 2.7. Dietary Inflammatory Index

The Dietary Inflammatory Index (DII) [3,4] was used as the reference in this research. This index is able to predict significant changes in C-reactive protein levels, assigning an inflammatory value for each nutrient (the more negative the value, the more anti-inflammatory the nutrient) [3].

Of the total nutrients for which there is information available regarding their inflammatory nature, 25 of the parameters initially included in the index have been considered in the present study. As a sample made up entirely of Spanish schoolchildren, the global averages and deviations of the amounts of nutrients consumed used in the original study [3,4] were replaced by values for Spanish schoolchildren taken from the ENALIA study [31,32], with the exception of the values for β-carotenes and fatty acids ω-3 which were taken from the global average [4].

### 2.8. Statistical Analyses 

Data analysis was performed using SPSS for Windows, 26.0. The Kolmogorov–Smirnov test was used to test continuum variable normality. All the analysed variables fitted normal distribution. Comparisons of two means in normal data were analysed using the t-test. Chi-square (χ^2^) was used to compare proportions. The significance level for all tests was set to *p* < 0.05.

#### 2.8.1. Random Forest (Decision Trees)

Decision trees are a data mining technique that predicts the value of a dependent variable based on the values of the independent variables. In this case, the CRT (Classification and Regression Trees) method was used, which allows the data to be classified into homogeneous groups with respect to the dependent variable and according to the criteria of the independent variables included in the analysis. The results are represented in a flow chart and allow us to predict the probability of belonging to each group of dependent variable, if the independent variables of a certain specific situation are known [33]. This procedure allows us to identify homogenous groups with respect to the dependent variable and facilitates predictions according to the independent variables included in the analysis. The rules are based on different methods and criteria. In the present analysis, the Gini criterion, also known as the Gini diversity index, was used. The Gini impurity measure at node t is defined as i(t) = 1 − S, where S (the impurity function) = ∑p 2 (j|t), for j = 1, 2, …, k. The impurity function attains a maximum if each class (or node) in the population occurs with equal probability, i.e., p(1|t) = p(2|t) = … = p(j|t). On the other hand, the impurity function attains its minimum (=0) if all cases at a given node belong to a single class. In other words, if node t is a pure node with a zero misclassification rate, then i(t) = 0.

A tree was generated, taking as the dependent variable the DII corrected by sex. The independent variables were age, father’s level of education (higher, secondary, middle, primary), mother’s level of education (higher, secondary, middle, primary), total screen time (min/week), and physical activity (min/week).

#### 2.8.2. Models of Logistic and Linear Regression

Five predictive models were created using multiple linear regression and logistic regression to assess which factors influence the presence of hypertension (NO/YES), CVRF (NO/YES), BFM%, and waist/height index. The Dietary Inflammatory Index adjusted by sex, the Dietary Inflammatory Index as a function of the 50th percentile (< or >P50) adjusted by sex, and the Kidmed score were included as independent variables.

## 3. Results

Table 1 describes the characteristics of the sample in terms of socio-economic status, habits and lifestyle, and anthropometric and cardiovascular health variables and indices for boys and girls. No significant differences were found by sex for any of the variables related to socio-economic background (area and parental level of education) and age. With regard to habits and lifestyles, the boys included in the sample have significantly longer overall screen time than girls, but there are no significant differences in physical activity. In terms of hemodynamic and nutritional status variables, significant differences between sexes were only observed in z-scores for size, weight, and body mass, which were significantly greater in boys than in girls. For the rest of the variables, no significant differences were found.

Table 2 shows the results of comparisons between boys and girls with regard to diet at the time of the study. 

Possible differences in energy and nutritional composition and quality of current diet between the sexes were also analysed. No statistically significant differences were found between boys and girls, except for cholesterol consumption, which was significantly higher in girls than in boys (Table 2). In terms of the quality of food, girls were found to consume better quality fat. By contrast, girls have a more pro-inflammatory diet than boys (Table 2).

As shown in Table 3, schoolchildren with a more pro-inflammatory diet (>P50) consume significantly less carbohydrates and fibre, consume poorer quality fat (lower ratio of polysaturated and monounsaturated fatty acid consumption), and have a poorer quality diet (lower Kidmed Index values) (Table 3).

Table 4 shows the results for differences between children below and above the 50th percentile for the DII, once corrected by sex for socioeconomic level, anthropometry, energy composition, and current diet. It was found that children above P50 for the DII, that is to say those who tend to consume a more pro-inflammatory diet, came from families with lower levels of education from both parents and currently had a significantly higher BMI than the WHO standard, higher body weight (BW), weight-height ratio (WHR), and BFM%. No significant differences were observed in the indicators of joint stiffness, but in the mean values for each variable there is a trend in the same direction as observed for the variables mentioned above.

Table 5 shows that children with a more anti-inflammatory diet (<P50) show a lower prevalence of hypertension and CVRF in general, with these differences being statistically significant.

Figure 1 shows a classification tree model made to predict whether boys and girls with certain characteristics are more likely to score higher values on the DII. The model fit is 58.2%, in other words, the model correctly classifies 58.2% of individuals. The dependent variable is the DII corrected by sex and grouped into two categories according to the P50 value, and the independent variables used in the analysis were age, mother’s level of education, father’s level of education, total screen time (min/week) and physical activity (min/week).

The root node (node 0) describes the dependent variable. The sample is then divided into two homogeneous subsets (nodes 1 and 2) determined by the father’s level of education as a predictor variable.

Node 2 groups children whose father has a primary, secondary, or middle school level of education, and this node has a higher prevalence of schoolchildren with a more pro-inflammatory diet (>P50) than node 1, which encompasses schoolchildren whose fathers have a higher level of education. Node 1 branches again based on total screen time (nodes 3 and 4). In node 4, children who spend more than 651.5 min/week in front of screens are more likely to have a pro-inflammatory diet (>P50) than in node 3 where children with screen time of less than 651.5 min/week are grouped.

Therefore, the nodes that define the profile of children with the highest prevalence of pro-inflammatory diet (*p* > 50) are nodes 2 and 4. There is a 61.2% probability of being above the mean if the father does not have a higher education (node 2). On the other hand, there is a 46.2% chance of being above average on the DII among children whose father has a higher education, but they spend more than 651.5 min/week in front of screens. 

The variables age, mother’s level of education, and physical activity (min/week) that were also introduced in the model were not significant and therefore the predictive model did not include them.

Table 6 shows predictive models for the indicator variables fat accumulation (BFM% and Waist-height ratio) and the presence of CVRF (obesity and hypertension, together) in childhood. Individuals with a pro-inflammatory diet (above the median for the DII) are 2.1 times more likely to develop hypertension in childhood (OR = 2.085) and have more than twice the risk (2.3) of developing obesity and hypertension simultaneously (of having these CVRF) (OR = 1.290) (Table 6). Furthermore, boys and girls with higher scores on the index, with a pro-inflammatory diet (>P50), will have higher values for BFM% and Waist/Height Index currently (Table 6). The total DII and Kidmed scores were not significant in any of the 5 models.

## 4. Discussion

In the present study, we used the dietary inflammatory index (DII) score to assess the capacity of overall dietary pattern to promote inflammation. Higher values on the DII represent a higher inflammatory potential of the diet. As expected, we observed that the DII was inversely associated with the intake of healthy nutrients, quality of ingested fat and adherence to the Mediterranean diet (Kidmed). Several studies have shown an inverse association between healthy diets and inflammation markers, as well as a direct association with “Western-like” dietary patterns [8,34]. The association between the DII and KIDMED index, as measures of food quality, is rather interesting because the group that scored below the median on the DII and therefore had a more anti-inflammatory diet has on average a higher KIDMED score. This confirms the relationship between these two indices that are indicators of high-quality dietary patterns [34].

Specifically, as in our sample, studies have related the consumption of fibre and cholesterol to low levels of inflammation markers. An anti-inflammatory diet is associated with high levels of anti-inflammatory markers (interleukins (IL-10) and tumour necrosis factor beta (TNF-b)) (IL-10 and TNF-b) found mainly in vegetables, fruit, fish, and legumes, i.e., foods that are higher in unsaturated fatty acids, fibre, vitamin C and micronutrients such as Calcium, Phosphorus, and Magnesium [12,34]. Although the molecular mechanisms of the association between these nutrients and their anti-inflammatory effect are not well known, there is evidence of the relationship between unsaturated fatty acids and a lower activation and even regulation of inflammatory gene expression and circulating biomarkers, meaning, therefore, there is no production of pro-inflammatory cytokines [35]. Thus, a single meal enriched with SFA is able to induce the IL-1 expression sharply and, if consumed regularly, it could trigger systemic inflammation, while increased consumption of unsaturated FA and fibre could moderate inflammation [35].

Differences between the sexes regarding total DII value could be related to higher cholesterol consumption and greater contribution of lipids in general to total dietary energy seen in girls. These results coincide with the data obtained for the DII in the original study, where significant differences between sexes were also found, and in particular, men also have a more anti-inflammatory diet [3]. In other words, even though it includes the 45 predictors of the DII from the original study, the DII used is a good indicator of inflammation in the diet for our school population.

Father’s level of education and total screen time are the variables that most determine the likelihood of having a pro-inflammatory diet. Previous studies have linked socioeconomic differences as a determinant of diet quality, with higher socio-economic levels determining better quality of diet [36]. Among Spanish children and adolescents, unhealthy eating shows a clear socioeconomic pattern: children and adolescents from households with higher socioeconomic levels have healthier consumption patterns [37]. The price of food is an important factor in its choice, and healthy foods are generally more expensive than unhealthy foods [36]. The level of parental education can be considered a good indicator of the socio-economic level of the family and, as in other studies, its relationship with the quality of food is noted [8]. There are few studies on the relationship between the DII and socioeconomic status, particularly in the Spanish population, so these results are especially relevant. The European IDEFICTS study found that dietary patterns in paediatric populations are influenced by parental socioeconomic status (SES), so children whose parents have a higher level of education and higher incomes have healthier dietary patterns, tending to consume less fast food and sugary products [38,39]. Other studies have shown that people with a lower SES tend to have lower-quality diets with high energy-dense food intake and lower fruit and vegetable consumption, which favours inflammation [38,39,40].

Furthermore, sedentary habits associated with excessive use of screens have been associated with a higher prevalence of overweight and obesity and inflammatory biomarkers in children [8,41]. Our results indicate that a more pro-inflammatory diet is related to longer screen time, but not to physical activity (hours a week). However, the time spent using screens could be spent on activities with higher energy expenditure [42,43], and ultimately this depends on families, their level of education, and socioeconomic status, as extracurricular physical activities are an additional expense for families. 

### Measures of Obesity and Dietary Inflammation

A pro-inflammatory DII was directly associated with the indices for general (BMI z-score, BFM (%) and FMI (kg/m2)) and abdominal obesity (WC (cm), Waist-height ratio), hypertension and obesity, for both sexes. These findings, like those noted in other studies [44,45], indicate the negative impact of a pro-inflammatory diet on preclinical cardiovascular phenotypes that can accumulate and become detectable during middle age, and presumably continue to accumulate thereafter [44,45]. Thus, cohort studies such as the one conducted by Navarro et al. [8], found an association between higher DII scores and an increased risk of overweight and obesity at 9 years of age (OR 1.06; 95% CI—1.01–1.09 and OR—1.12; 95% CI—1.07–1.18, respectively). A cross-sectional study of Iranian children aged 6 to 18 also found that subjects with a higher DII (>1.5) had higher BMI and BW values than those with a more anti-inflammatory diet [46]. Recent studies in Spain coincide with our results; a cross-sectional study conducted with schoolchildren aged 9 to 17 years showed a positive association between a pro-inflammatory diet and higher waist fat content measured by means of the WHR (β 0.128; 95% CI 0.001–0.16) [44]. 

The WHR is an indicator of increased cardiometabolic risk, and previous studies confirm the relationship between higher WHR and more pro-inflammatory values on the DII, and support the influence of a pro-inflammatory diet on excess weight and the development of obesity during childhood and adolescence [44]. On the other hand, our results also indicate that the group of children with a more anti-inflammatory diet has significantly lower prevalence of CVRF and hypertension. In relation to hypertension, studies in children are more limited and less conclusive. In the study conducted on adolescents by Sethna et al. [47], a positive association was observed between higher DII and SBP (β 5.07, 95% CI 2.55–7.59) and a negative association with lower DBP (β 4.14, 95% CI 6.74, 1.54). In addition, it should be borne in mind that pre-existing information studying the relationship between DII and hypertension and CVRF in general is conditioned by the fact that most of these studies focus on risk groups [45]. This increases the value of these results as this sample does not include risk groups. Overall, the results reinforce the idea that promoting an anti-inflammatory diet is a useful prevention strategy in reducing the risk of CVD and improving long-term health [44]. 

Finally, the work presented here is novel in the sense that it introduces an adaptation of the DII with reference values for Spanish schoolchildren and furthers explores the relationship between the inflammatory nature of diet, evaluated by means of the DII, and cardiovascular risk factors, controlling for the effect of differences in socio-economic status, lifestyle, and anthropometric indicators. However, several limitations should be considered. First, the design of the study was cross-sectional, and the interpretation of findings should be made cautiously. Second, BP values were measured only on two occasions in all included surveys, and the definition of pediatric hypertension should be based on BP measurements on at least three different occasions (27). Nevertheless, the results obtained reveal an association between the inflammatory properties of diet and socioeconomic status, various anthropometric indicators, hypertension, and CVRF in general.

## 5. Conclusions

The inflammatory nature of diet is related to the socio-economic level of families and to children’s screen time: the lower the socio-economic level, the more pro-inflammatory the diet. 

The father’s level of education is the variable that best predicts the pro-inflammatory or anti-inflammatory character of a child’s diet. 

Among children whose fathers have a higher level of education, those who spend the most time playing games consoles, computers, watching television and in front of screens have higher values on the DII. 

The anti-inflammatory nature of the diet, measured by means of the DII, is positively associated with a higher KIDMED score, both of which are indicators of diet quality. 

Finally, children with a pro-inflammatory diet are more likely to develop obesity and hypertension, as well as to have greater body fat accumulation.

## Figures and Tables

**Figure 1 ijerph-19-15706-f001:**
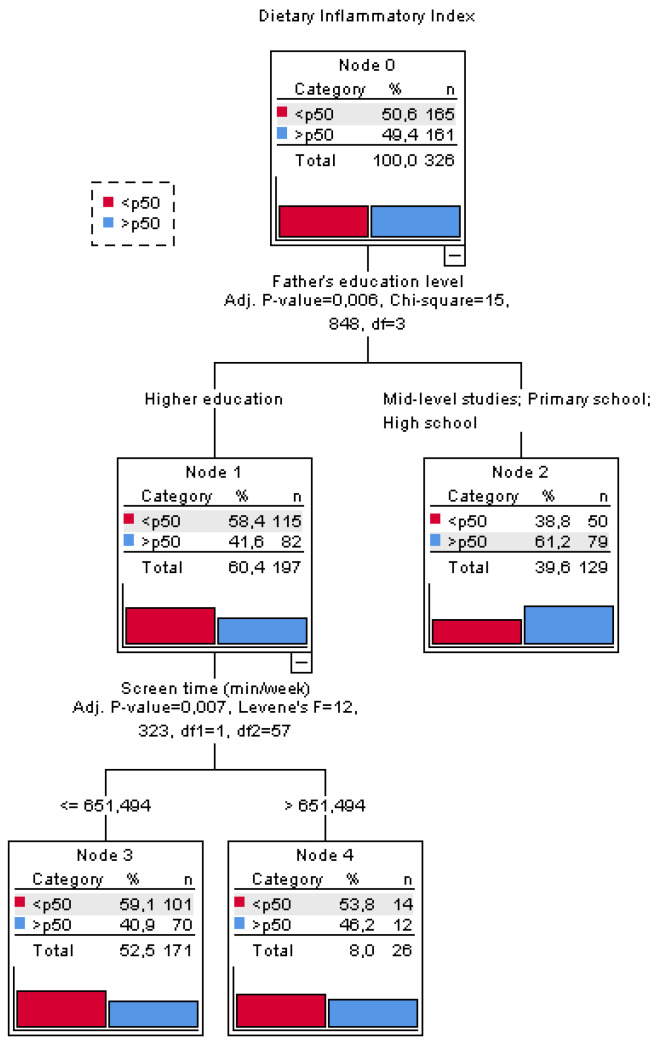
Decision tree based on the dietary inflammatory index above or below the median, father’s level of education (higher or secondary, middle, and primary), and total screen time below or above 651.5 min/week.

**Table 1 ijerph-19-15706-t001:** Socio-economic characteristics, nutritional status, WHO standard Z scores for size and age, and current hemodynamic variables by sex.

	Total		Male		Female		
	N	N(%)/ Mean (sd)	N	N(%)/ Mean (sd)	N	N(%)/ Mean (sd)	*p*-Value
Socio-economic characteristics							
Age (years)	365	9.6 (1.1)	190	9.7 (1.1)	175	9.6 (1.2)	0.267
Place of residence within Region of Madrid							
North	365	128 (35.1)	190	62 (32.6)	175	66 (37.7)	0.575
Centre		150 (41.1)		80 (42.1)		70 (40.0)	
South		87 (23.8)		48 (25.3)		39 (22.3)	
Mother’s level of education							
Primary school	304	21 (6.9)	156	13 (8.3)	148	8 (5.4)	0.364
High school		30 (9.9)		14 (9.0)		16 (10.8)	
Middle school		52 (17.1)		31 (19.9)		21 (14.2)	
Higher Education		201 (66.1)		98 (62.8)		103 (69.6)	
Father’s level of education							
Primary school	300	31 (10.3)	154	18 (11.7)	146	13 (8.9)	0.108
High school		34 (11.3)		15 (9.7)		19 (13.0)	
Middle School		49 (16.3)		32 (20.8)		17 (11.6)	
Higher Education		186 (62.0)		89 (57.8)		97 (66.4)	
Screen time (min/week)	145	466.8 (463.9)	72	604.4 (496.6)	73	331.1 (386.8)	**<0.001**
Physical activity (min/week)	144	425.2 (265.1)	51	444.9 (330.6)	65	406.0 (180.4)	0.385
Nutritional status and WHO standard Z scores for size, early school age							
Height (cm)	365	141.7 (9.0)	190	142.4 (8.1)	175	141.0 (9.9)	0.142
Weight (kg)	365	37.3 (9.0)	190	37.9 (8.8)	175	36.7 (9.2)	0.186
BMI (kg/m2)	365	18.4 (2.9)	190	18.5 (2.9)	175	18.2 (3.0)	0.335
WC(cm)	362	65.9 (8.9)	188	66.5 (8.9)	174	65.3 (8.9)	0.186
Fat mass (kg)	365	9.5 (5.1)	190	9.5 (4.9)	175	9.6 (5.3)	0.906
BFM (%)	365	24.1 (7.8)	190	23.8 (7.3)	175	24.5 (8.4)	0.340
FMI (kg/m^2^)	365	4.6 (2.2)	190	4.6 (2.1)	175	4.7 (2.3)	0.626
HAZ	365	0.4 (0.9)	190	0.5 (0.9)	175	0.2 (1.0)	**0.004**
WAZ	185	0.6 (1.1)	94	0.8 (1.1)	91	0.4 (1.1)	**0.012**
BMIZ	365	0.6 (1.1)	190	0.7 (1.1)	175	0.4 (1.1)	**0.034**
Obesity	365	41 (11.2)	190	24 (12.6)	175	17 (9.7)	0.643
Hemodynamic variables							
SBP (mmHg)	365	103.4 (15.1)	190	103.8 (15.9)	175	102.9 (14.2)	0.569
DBP (mmHg)	365	63.0 (10.9)	190	63.0 (11.3)	175	63.0 (10.5)	0.981
MAP (mmHg)	365	76.5 (10.9)	190	76.6 (11.4)	175	76.3 (10.4)	0.806
Hypertension	365	52 (14.2)	190	29 (15.3)	175	23 (13.1)	0.563

Abbreviations: BMI: Body Mass Index; WC: Waist circumference (cm); BFM%: Percentage of Body Fat Mass; FMI: Fat Mass Index; HAZ: height for age Z-score; WAZ: Weight for age Z-score; BMIZ: Body Mass Index for age Z-score; SBP: Systolic Blood Pressure; DBP: Diastolic Blood Pressure; MAP: Mean Arterial Pressure. Bold value are significant, *p* < 0.05.

**Table 2 ijerph-19-15706-t002:** Nutrient intake and diet quality by sex.

	Total		Boys		Girls		
	N	Mean (Std)	N	Mean (Std)	N	Mean (Std)	*p*-Value
Carbohydrate (% energy)	358	46.6 (5.5)	185	46.9 (5.4)	173	46.4 (5.6)	0.421
Total Fat (% energy)	358	36.6 (5.2)	185	36.4 (5.2)	173	36.7 (5.2)	0.641
Protein (% energy)	358	16.4 (2.5)	185	16.4 (2.7)	173	16.5 (2.3)	0.826
PUFA (% energy)	353	4.4 (1.3)	182	4.5 (1.4)	171	4.3 (1.0)	0.362
SFA (% energy)	353	13.4 (2.9)	182	13.6 (3.3)	171	13.2 (2.6)	0.121
MUFA (% energy)	353	16.0 (2.9)	182	15.8 (2.9)	171	16.1 (2.8)	0.291
PUFA + MUFA/SFA	358	1.6 (0.3)	185	1.5 (0.3)	173	1.6 (0.3)	**0.036**
Fibre (mg)/1000 kcal	358	9.5 (2.9)	185	9.4 (2.6)	173	9.7 (3.2)	0.246
Cholesterol (mg)/1000 kcal	358	149.8 (43.1)	185	144.9 (41.4)	173	155.0 (44.3)	**0.027**
KIDMED Index	347	6.6 (1.7)	178	6.5 (1.7)	169	6.8 (1.7)	0.076
DII	352	1.5 (5.6)	182	0.9 (5.6)	170	2.1 (5.4)	**0.038**

Abbreviations: PUFA: Polyunsaturated fatty acid; SFA: Saturated fatty acid; MUFA: Monounsaturated fatty acid; DII: Dietary Inflammatory Index. Bold value are significant, *p* < 0.05.

**Table 3 ijerph-19-15706-t003:** Differences in nutrient intake and diet quality among subjects with or without a dietary inflammatory index above the 50th percentile.

	Dietary Inflammatory Index *			
	N (%)/Mean (sd)			
	Total	<P50	>P50	*p*-Value
Current intakes				
Energy intake (Kcal/d)	2267.1 (406.8)	2331.2 (602.9)	1805.6 (369.7)	**<0.001**
Carbohydrate (% energy)	46.6 (5.5)	47.8 (5.2)	45.4 (5.7)	**<0.001**
Total Fat (% energy)	36.6 (5.2)	35.2 (4.9)	38.1 (5.1)	**<0.001**
Protein (% energy)	16.4 (2.5)	16.7 (2.2)	16.2 (2.7)	0.077
PUFA (% energy)	16.0 (2.9)	15.4 (2.5)	16.5 (3.0)	**<0.001**
SFA (% energy)	4.4 (1.3)	4.5 (1.4)	4.4 (1.2)	0.365
MUFA (% energy)	13.4 (2.9)	12.6 (2.6)	14.2 (3.0)	**<0.001**
PUFA + MUFA/SFA	1.6 (0.3)	1.6 (0.3)	1.5 (0.3)	**0.001**
Fibre (mg)/1000 kcal	9.5 (2.9)	10.2 (2.7)	8.7 (2.0)	**<0.001**
Cholesterol (mg)/1000 kcal	149.8 (43.1)	143.2 (40.6)	155.7 (44.6)	**0.006**
Kidmed Index		6.94(1.48)	6.31(1.79)	**0.001**

Abbreviations: PUFA: Polyunsaturated fatty acid; SFA: Saturated fatty acid; MUFA: Monounsaturated fatty acid. * corrected by sex. Bold value are significant, *p* < 0.05.

**Table 4 ijerph-19-15706-t004:** Differences in socioeconomic, anthropometric, and hemodynamic characteristics between subjects with or a without a Dietary Inflammatory Index above the 50th percentile.

		Dietary Inflammatory Index *			
		N (%)/Mean (sd)			
		Total	<P50	>P50	*p*-Value
DII *		1.496 (5.549)	−2.355 (5.257)	5.347 (2.074)	**<0.001**
Age (years)		9.6 (1.1)	9.7 (1.1)	9.6 (1.2)	0.310
Sex (Female)		175 (47.9)	85 (48.3)	85 (48.3)	0.542
Place of residence within Region of Madrid					
	North	128 (35.1)	65 (36.9)	56 (31.8)	0.135
	Centre	150 (41.1)	76 (43.2)	69 (39.2)	
	South	87 (23.8)	35 (19.9)	51 (29.0)	
Mother’s level of education					
	Primary school	21 (6.9)	5 (3.3)	16 (11.0)	**0.030**
	High school	30 (9.9)	12 (8.0)	18 (12.3)	
	Middle school	52 (17.1)	25 (16.7)	24 (16.4)	
	Higher Education	201 (66.1)	108 (72.0)	88 (60.3)	
Father’s level of education					
	Primary school	31 (10.3)	11 (7.4)	18 (12.3)	**0.001**
	High school	34 (11.3)	15 (10.1)	18 (12.3)	
	Middle school	49 (16.3)	14 (9.5)	34 (23.3)	
	Higher Education	186 (62.0)	108 (73.0)	76 (52.1)	
Screen time (min/week)		466.8 (463.9)	435.7 (441.1)	480.1 (481.6)	0.579
Physical activity (min/week)		425.2 (265.1)	414.4 (175.5)	454.9 (323.8)	0.387
HAZ		0.4 (0.9)	0.3 (0.9)	0.4 (0.9)	0.491
WAZ		0.6 (1.1)	0.5 (1.0)	0.8 (1.1)	0.054
BMIZ		0.6 (1.1)	0.4 (1.1)	0.7 (1.1)	**0.017**
WC (cm)		65.9 (8.9)	79.6 (9.5)	80.2 (10.5)	**0.034**
Waist-height ratio		0.5 (0.1)	0.5 (0.05)	0.5 (0.1)	**0.004**
Fat mass (kg)		9.5 (5.1)	9.1 (4.6)	10.1 (5.4)	0.051
BFM (%)		24.1 (7.8)	23.3 (7.4)	25.1 (8.2)	**0.027**
FMI (kg/m^2^)		4.6 (2.2)	4.4 (2.0)	4.9 (2.4)	**0.017**
SBP (mmHg)		103.4 (15.1)	102.7 (14.4)	104.1 (15.9)	0.408
DBP (mmHg)		63.0 (10.9)	61.8 (11.0)	63.9 (10.8)	0.064
MAP (mmHg)		76.5 (10.9)	75.4 (10.8)	77.3 (11.1)	0.108

Abbreviations: BMI: Body Mass Index; WC: Waist circumference (cm); BFM%: Percentage of Body Fat Mass; FMI: Fat Mass Index; HAZ: height for age Z-score; WAZ: Weight for age Z-score; BMIZ: Body Mass Index for age Z-score; SBP: Systolic Blood Pressure; DBP: Diastolic Blood Pressure; MAP: Mean Pressure Arterial. DII: Dietary Inflammatory Index. * corrected by sex. Bold value are significant, *p* < 0.05.

**Table 5 ijerph-19-15706-t005:** Dietary Inflammatory Index relation with a prevalence of Cardiovascular Risk Factors.

		Dietary Inflammatory Index *			
		Total N (%)	<P50 N (%)	>P50 N (%)	*p*-Value
BMI	Normal weight	229 (62.7)	115 (65.3)	104 (59.1)	0.082
	Overweight	95 (26.0)	48 (27.3)	46 (26.1)	
	Obesity	41 (11.2)	13 (7.4)	26 (14.8)	
Hypertension	Yes	52 (14.2)	18 (10.2)	32 (18.2)	**0.047**
CVRF	No CVRF	288 (78.9)	151 (85.8)	128 (72.7)	**0.023**
	Hypertension	36 (9.9)	12 (6.8)	22 (12.5)	
	Obesity	25 (6.8)	7 (4.0)	16 (9.1)	
	Hypertension and obesity	16 (4.4)	6 (3.4)	10 (5.7)	

Abbreviations: BMI: Body Mass Index; CVRF: presence of Cardiovascular Risk Factors. * corrected by sex. Bold value are significant, *p* < 0.05.

**Table 6 ijerph-19-15706-t006:** Associations between the Dietary Inflammatory Index, nutritional status, and cardiovascular risk factors in children.

	β	*p*	OR	CI 95%
Model 1 ^†^: Hypertension (No/Yes)				
Dietary Inflammatory Index (<p50/>p50) *	0.735	**0.023**	**2.085**	**(1.107–3.927)**
Dietary Inflammatory Index *	−0.220	0.386	0.802	(0.487–1.320)
KIDMED Index	0.086	0.357	1.090	(0.908–1.308)
Dietary Inflammatory Index (<p50/>p50) *	0.829	**0.003**	**2.290**	**(1.316–3.985)**
Dietary Inflammatory Index *	−0.068	0.759	0.934	(0.605–1.442)
KIDMED Index	0.064	0.431	1.066	(0.909–1.251)
Dietary Inflammatory Index (<p50/>p50) *	1.957	**0.026**	-	0.240–3.674
Dietary Inflammatory Index *	0.578	0.403	-	−0.792–1.965
KIDMED Index	0.054	0.836	-	−0.463–0.507
Dietary Inflammatory Index (<p50/>p50) *	0.015	**0.012**	-	0.003–0.027
Dietary Inflammatory Index *	−0.004	0.461	-	−0.013–0.006
KIDMED Index	0.000	0.929	-	−0.004–0.003

Abbreviations: CVRF: presence of Cardiovascular Risk Factors; BFM: Body Fat Mass (%); * corrected by sex. † Logistic regression was used. Model 1: R^2^ = 0.034; *p*-value = 0.091. Model 2: R^2^ = 0.042 *p*-value = 0.027. Dependent variables: Model 1: Hypertension, Model 2: Presence of CVRF. Independent variables: Dietary Inflammatory Index (>P50), corrected by sex, total Dietary Inflammatory Index corrected by sex, and Kidmed index. Model 3: R^2^ = 0.008; *p*-value = 0.132. Model 4: R^2^ = 0.013; *p*-value = 0.062. Dependent variables: Model 3: % BFM%, Model 4: Waist/Height Index. Independent variables: Dietary Inflammatory Index (>P50), corrected by sex, total Dietary Inflammatory Index corrected by sex, and Kidmed index.

## Data Availability

The datasets used and/or analysed during the current study is available from the corresponding author on reasonable request.

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
