# Peer review of "Relationship between the Dietary Inflammatory Index and Cardiovascular Health among Children"

_ijerph, 2022, doi:10.3390/ijerph192315706_

Round 1

Reviewer 1 Report

Dear authors,

The present article evaluates satisfactorily the relationship between inflammation generated

by diet and cardiovascular risk in a children population, considering aspects such as

socioeconomic level, lifestyles and anthropometric parameters. The inclusion of these items in

the analysis of said association is a novel point with respect to previous articles and allows a

better understanding of the possible interactions, which makes this research relevant.

Next, I detail some aspects of your manuscript that I consider should be corrected or better

explained:

The company that produces the equipment used as well as its place of production must be

indicated in the methodology section.

Line 154: When it refers to the consumption of cholesterol per 1000 kcal, it should be

indicated in this way in the units because, according to what is presented in the text, it could

be interpreted as referring to mg per kcal.

Table 1: some decimal points are missing in the p-values

Paragraph beginning on line 216: reference is made to table 2 and a comment is also made

regarding the weeks of breastfeeding, although this information does not appear in the table.

It is advisable to include this information in the table or put this text in a different paragraph.

Tables 2 and 3: There are abbreviations in footnote that are not found in the Tables.

Table 3: Breastfeeding patterns are alluded to in the title of the Table, but this information is

not found in the table.

Paragraph beginning on line 229: reference is made to table 4 but some of the variables

specified in that paragraph are found in table 3 instead of 4.

Line 234: the abbreviations BW and WHR appear for the first time, but the full term to which it

refers is not included. Data on BW do not appear in the tables, so perhaps they should be

included in the text.

Lines 256, 258, 263 and footnote to Figure 1: reference is made to the time of use of screens

that divides the sample into two parts. However, the indicated value does not coincide every

time, so it would be necessary to review it to determine which is the correct value.

In Tables 4 and 6 and in lines 273 and 341, reference is made to the weight/height ratio, while

in line 279 and in footnote to Table 6, reference is made to the waist/height index. Check if the

same index is referenced every time and correct it if necessary.

Kind regards.

Author Response

Response to Reviewer 1 Comments

Thank you for your review and interesting comments.

We will try to respond to your comments below:

The company that produces the equipment used as well as its place of production must be indicated in the methodology section.

Response:

Amended in text

Line 154: When it refers to the consumption of cholesterol per 1000 kcal, it should be indicated in this way in the units because, according to what is presented in the text, it could be interpreted as referring to mg per kcal.

Response:

Amended in text

Table 1: some decimal points are missing in the p-values

Response:

Amended in Table 1

Paragraph beginning on line 216: reference is made to table 2 and a comment is also made regarding the weeks of breastfeeding, although this information does not appear in the table. It is advisable to include this information in the table or put this text in a different paragraph.

Response:

It has been removed from the text because the study focuses on the children's eating habits at the time they were measured and not on breastfeeding patterns.

Tables 2 and 3: There are abbreviations in footnote that are not found in the Tables.

Response:

Amended in Tables 2 and 3

Table 3: Breastfeeding patterns are alluded to in the title of the Table, but this information is not found in the table.

Response:

Amended in Table 3

Paragraph beginning on line 229: reference is made to table 4 but some of the variables specified in that paragraph are found in table 3 instead of 4.

Response:

The information presented in said paragraph has been revised and it only refers to the information contained in table 4. Therefore, it has been kept as it was.

Line 234: the abbreviations BW and WHR appear for the first time, but the full term to which it refers is not included. Data on BW do not appear in the tables, so perhaps they should be included in the text.

Response:

Terms referred to in the abbreviations have been included. Data related to BW was included in the table represented by the Weight for age Z-score

Lines 256, 258, 263 and footnote to Figure 1: reference is made to the time of use of screens that divides the sample into two parts. However, the indicated value does not coincide every time, so it would be necessary to review it to determine which is the correct value.

Response:

Amended in text

En los cuadros 4 y 6 y en las líneas 273 y 341 se hace referencia a la relación peso/altura, mientras que en la línea 279 y en la nota al pie del cuadro 6 se hace referencia al índice cintura/altura. Compruebe si se hace referencia al mismo índice cada vez y corríjalo si es necesario.

Respuesta:

Texto modificado

Reviewer 2 Report

The article "Relationship between the dietary inflammatory index and cardiovascular health among children" is interesting and well written. However, there are some aspect that could be improved. 

First of all, it is not clear for me if the Dietary Inflammatory Index was validated for the entire population, only for children or only for adults. This creates some confusion even in discussions part. Then, are the levels of inflammatory markers equal for each age? Or do they change according to age and sex?

Other little aspect below. 

Abstract. There is not background, but only the explanation of the study aim. Please add something. 

Introduction. A reference is needed in line 36-38. Please explain why did you choose to study this part of population (children), becuae it is not so clear. 

Material and methods.  Why did you exluse girls with menarche? Lines 135-138, did you refer to children with hypertension that did not use medicines, right? Line 150, please explain the acronyms. 

Results. Please homogenize the tables.

Conclusions. Lines 285-287, I think some references are missing. You wrote "several studies" nd then there is only a reference. 

Author Response

Response to Reviewer 2 Comments

Thank you for your review and interesting comments.

We will try to respond to your comments below:

First of all, it is not clear for me if the Dietary Inflammatory Index was validated for the entire population, only for children or only for adults. This creates some confusion even in discussions part. Then, are the levels of inflammatory markers equal for each age? Or do they change according to age and sex?

Response:

The Dietary Inflammatory Index has been used in both adults and children as can be seen in the following reference. Hébert JR, Shivappa N, Wirth MD, Hussey JR, Hurley TG. Perspective: The Dietary Inflammatory Index (DII)-Lessons Learned, Improvements Made, and Future Directions. Adv Nutr. 2019 Mar 1;10(2):185-195. doi: 10.1093/advances/nmy071. PMID: 30615051; PMCID: PMC6416047.

This article shows the results of the values of the index, so levels of inflammatory markers have not been presented or interpreted.

Abstract. There is not background, but only the explanation of the study aim. Please add something.

Response:

Abstract is limited to a maximum of 200 words, currently having a total of 197. Therefore, there is no space to expand the background in this section.

Introduction. A reference is needed in line 36-38. Please explain why did you choose to study this part of population (children), becuae it is not so clear.

Response:

References have been included in lines 36-38.

Children were selected for this research due to the importance of the life cycle approach for the study of diseases. Different studies have revealed the important relationship between the different stages of life and the development of diseases in adulthood.

Material and methods.  Why did you exluse girls with menarche? Lines 135-138, did you refer to children with hypertension that did not use medicines, right? Line 150, please explain the acronyms.

Response:

Girls with menarche were excluded due to changes in body composition.

Children with hypertension did not know their condition. The children included in the study had no history of heart disease and hypertension in terms of their parents and themselves.

Explanation of acronyms has been included in line 150.

Results. Please homogenize the tables.

Response:

We do not understand what you mean in this comment. If it is necessary to make any change, please, specify it clearly.

Conclusiones. Líneas 285-287, creo que faltan algunas referencias. Usted escribió "varios estudios" y luego solo hay una referencia.

Respuesta:

Se ha incluido la referencia faltante

Reviewer 3 Report

Minor revision

This article reports the results of an interesting study,  whilst controlling for socio-economic status and lifestyle
factors (physical activity and screen time), we assess the relationships between nutrient
intake or food group consumption and the DII, as well as the association between the DII
and cardiovascular health indicators (central and abdominal obesity and blood pressure)
in a sample of 8-12-year-old schoolchildren living in the Region of Madrid.

This study provides results on a significant sample of the autonomous community of Madrid (n = 365 schoolchildren). This is considered a strength, as it represents a great effort, statistically significant results, as well as a poorly studied population. In addition, the manuscript is structured and properly written, its design is correct, and the statistical analysis seems adequate. The authors should, however, clarify information on some of the topics presented below:

In the keywords I would add KIDMED and 24-hour recall surveys.

In the objectives, the age of the sample must be corrected since children from 8 to 11 years old appear instead of 8 to 12 years old.

In the method section, it only describes that the questionnaire is composed of 16 items , it is necessary to define which are the items.

In the results section, the main findings are clearly described, and the tables improve the understanding of the contents.

In the discussion, an adequate comparison is made with studies with similar characteristics. This including the conclusion with the main findings, as well as the importance of of promoting an anti-inflammatory diet

The bibliography consulted is relevant.

Author Response

Response to Reviewer 3 Comments

Thank you for your review and interesting comments.

We will try to respond to your comments below:

In the keywords I would add KIDMED and 24-hour recall surveys.

Response:

Added in text

In the objectives, the age of the sample must be corrected since children from 8 to 11 years old appear instead of 8 to 12 years old.

Response:

Amended in text

In the method section, it only describes that the questionnaire is composed of 16 items, it is necessary to define which are the items.

Response:

The questionnaire includes the items of the kidmed index, which is referenced in the current line 144. In order not to extend the paper, the specific items were not included as they are detailed in said reference (reference 28), but if considered appropriate, they could be included.

In the results section, the main findings are clearly described, and the tables improve the understanding of the contents.

En la discusión, se hace una comparación adecuada con estudios con características similares. Esto incluye la conclusión con los principales hallazgos, así como la importancia de promover una dieta antiinflamatoria

La bibliografía consultada es relevante.

Respuesta:

Muchas gracias por sus comentarios.

Reviewer 4 Report

Unfortunately, this article fails to explain how the Dietary Inflammatory Index(DII) has been adapted from an adult's study(reference 4) to a children's study. Moreover, the authors refer to CRP as the target (lines 157-160) for this index when in fact it is related to hs-CRP which definitely is not the same and is the correct biomarker that correlates with chronic inflammation in obesity. Furthermore, they don't explain how they adapted the DII with the specified nutrients in the article, and more importantly, they do not present any hs-PCR data to validate their findings. This should have been the key point of their findings. Last but not least they neither explain the performance of their Random Forest models nor they address the limitations of this study.

Author Response

Response to Reviewer 4 Comments

Thank you for your review and interesting comments.

We will try to respond to your comments below:

Unfortunately, this article fails to explain how the Dietary Inflammatory Index(DII) has been adapted from an adult's study(reference 4) to a children's study. Moreover, the authors refer to CRP as the target (lines 157-160) for this index when in fact it is related to hs-CRP which definitely is not the same and is the correct biomarker that correlates with chronic inflammation in obesity. Furthermore, they don't explain how they adapted the DII with the specified nutrients in the article, and more importantly, they do not present any hs-PCR data to validate their findings.

Response:

As it says in the introduction, reference is made to hs-CRP, making it clear that it is the one that has been examined in the index used. The text also includes the references of said index in which it is explained how the index was created.

In this reference, among others, it can be seen research that includes children and uses this same index. Hébert JR, Shivappa N, Wirth MD, Hussey JR, Hurley TG. Perspective: The Dietary Inflammatory Index (DII)-Lessons Learned, Improvements Made, and Future Directions. Adv Nutr. 2019 Mar 1;10(2):185-195. doi: 10.1093/advances/nmy071. PMID: 30615051; PMCID: PMC6416047.

Last but not least they neither explain the performance of their Random Forest models nor they address the limitations of this study.

Response:

The performance of the Random Forest models is included in the line 243-244.

Reference to the limitations of the study is made in the last two paragraphs of the discussion.

Round 2

Reviewer 2 Report

Thank you to the author to have improved they article. However, there are still some aspect to fix. 

Thank you for the explanation, but I think to have to add it in the paper, because if some aspects were not clear for me it could be the same for readers. So, please add with you choose to analyze children and why did you exluded girls with menarche. 

There are too many lines inside the table. Why did you not put all the parameters togheter from height to obesity and from SBP to Hypertension?

Author Response

Thank you for your review and interesting comments.

We will try to respond to your comments below:

Thank you for the explanation, but I think to have to add it in the paper, because if some aspects were not clear for me it could be the same for readers. So, please add with you choose to analyze children and why did you exluded girls with menarche. 
- Included in the introduction and the material and methods.

There are too many lines inside the table. Why did you not put all the parameters togheter from height to obesity and from SBP to Hypertension?

- Modified in tables